# The Thermal Behavior of γ-PA1010: Evolution of Structure and Morphology in the Simultaneous Thermal Stretched Films

**DOI:** 10.3390/ma13071722

**Published:** 2020-04-07

**Authors:** Zhenya Zhang, Wentao Liu, Hao Liu, Aihua Sun, Yeonwoo Yoo, Suqin He, Chengshen Zhu, Mingcheng Yang

**Affiliations:** 1School of Materials Science and Engineering, Zhengzhou University, No.100 Science Avenue, Zhengzhou 450001, China; zhangzhenya0924@126.com (Z.Z.); hliu@zzu.edu.cn (H.L.); hesq@zzu.edu.cn (S.H.); zhucs@zzu.edu.cn (C.Z.); 2Isotope Institute Co., Ltd., Henan Academy of Sciences, 7 Songshan South Road, Zhengzhou 450015, China; ymch-7305@126.com; 3Key Laboratory of Additive Manufacturing Material of Zhejiang Province, Department of Functional Materials and Nano-Devices, Ningbo Institute of Materials Technology & Engineering, Chinese Academy of Sciences, 1219 Zhongguan West Road, Ningbo 315201, China; sunaihua@nimte.ac.cn; 4Department of Coating Technology, Korea Institute of Materials Science, 797 Changwondaero, Changwon 51508, Korea; yooyw08@kims.re.kr

**Keywords:** PA1010, crystallinity, orientation, draw ratio, simultaneous thermal stretched

## Abstract

In this work, polyamide 1010 (PA1010) films were prepared by melt-quenching. A wide-angle X-ray diffractometer (WAXD) with a thermal stretching stage was used to investigate the structure transformation, crystallinity and degree of orientation in the course of simultaneous thermally stretched PA1010. The crystallinity increased along with the increase of draw ratio and then decreased as the draw ratio was over 2.00 times—which the maximum value reached when the draw ratio was about 2.00 times. The degree of orientation of γ-PA1010 was much greater at higher temperature than room temperature (RT); the difference gradually became weaker with the increase of draw ratio. There was a linear relationship between the draw ratios and tensile force at higher temperatures, and the tensile force increased with the increase of draw ratios. The tensile force may induce crystallization and promote orientation in the course of simultaneous thermally stretched PA1010. These phenomena are beneficial to understand the structure-processing-performance relationship and provide some theoretical basis for the processing and production.

## 1. Introduction

Polyamide (PA), an important semi-crystalline polymer, is used in many fields, such as military equipment [1], aerospace materials [2], insulation and textile materials [3,4], due to its good mechanical, thermal and chemical properties [5,6]. The crystal structure and structural transformation play decisive roles in the properties of the materials during processing [7,8]. Some researchers have extensively explored the crystalline phase transitions during the process of heating or cooling [9,10,11]. The triclinic α-crystal structure of PA 66 may convert into the pseudo γ-hexagonal-crystal structure in the course of heating, in a process named the Brill transition [12,13,14]; the corresponding transition temperature is named the Brill transition temperature, or simply *T_B_* [15,16,17,18,19]. Generally speaking, the *T_B_* will be changed with the grain sizes and state, and sometimes even exceeds the melting point (*T_m_*) [20,21]. The *T_B_* of the polymer decreases as the number of methylene groups increases in the polymer molecular chain [22,23]. To date, most of the research hot topics have mainly focused on short-chain polyamides such as PA56, PA66, PA49 and PA12 etc. Research focused on long-chain polyamides is still scarce [24,25,26,27,28].

PA1010 was independently developed by China and commercialized in 1961. PA1010 is widely used in industrial silk, civilian silk, rods and other fields, which also has the characteristics of low density, low water absorption and chemical stability [28]. There are two basic crystal forms for PA1010: triclinic α-crystal and pseudo γ-hexagonal crystal. The γ-crystal is the base crystal structure and could exist stably at room temperature [29]. The α-crystal of PA1010 is a research hot topic and has been concerned widely because α-crystal could transform to γ-crystal structure when the temperature is over 100 °C. In recent decades, Zhu and Mo [30,31,32,33] have studied the condensed matter structure of PA1010 and clarified the structure of PA1010 in detail. Tashiro et al. [34,35,36,37] have reported the PA1010 Brill transition and characterized some group changes with Fourier transform infrared spectroscopy (FT-IR). Wang et al. [38] have reported structural changes of PA1010 when PA1010 was stretched at room temperature. We et al. [39] have reported the structural evolution of PA1010 with the α-crystal under the simultaneous thermal stretching and confirmed the tensile force and higher annealing temperature could affect the crystallinity and degree of orientation. Although many researchers have known the reports on the crystal formation of PA1010 with the α-crystal, the researches about the dynamic structural evolution and properties of PA1010 with the γ-crystal under the simultaneous thermal stretching are rarely reported, especially for the study on the orientation.

In this work, the structure variation of PA1010 are systematically studied with in situ WAXD, The PA1010 with the γ-crystal was prepared by melt-quenched. And the effects of the higher annealing temperatures and tensile force on structure evolution, crystallinity and orientation of γ-PA1010 were investigated.

## 2. Experimental

PA1010 is purchased from Zhenwei Composite Materials Co., Ltd., Shanghai, China. The average molecular weight of PA1010 is equal to 1.42 × 10^4^ and the melting point is 203 °C. The PA1010 was dried at 100 °C for 8 h to eliminate moisture. It was wrapped with polyimide and then melted at 230 °C with 10 MPa for 5 min in a vacuum laminator (Beijing Future Material Sci-tech Co., Ltd., Beijing, China), then moved into ice-water mixture to quench quickly. Dimension of the sample was cut into 24 mm × 4 mm × 2 mm.

The structure evolution of all specimens was measured by in situ WAXD (BRUKER AXS GMBH, Karlsruhe, Germany) (D8 DISCOVER, Cu-Kα radiation, 2 Å), which was equipped with the hot-stretching stage. The specimens were respectively annealed at 100, 110, 120, 130 and 140 °C; the simultaneously stretched velocity was 10 μm/s. The WAXD data collected at the draw ratios were 1.25, 1.50 1.75, 2.00, 2.25, 2.50, 2.75 and 3.00 times, respectively.

The crystallinity (*W_c,x_*) was calculated according to Equation (1), on the basis of area of diffraction spots (crystalline phase) and area of amorphous phase scattering. Some researchers [28,30,31] replaced the crystal area with crystal strength and obtained the areas of crystal region and amorphous region with peak fitting software [40,41].
(1)$$W_{cx}=\frac{A_{c}}{A_{c}+A_{a}}$$
where the *A_c_* and *A_a_* was the crystal and amorphous area, respectively.

The degree of orientation (Π) was obtained by Equation (2) [40,41],
(2)$$\Pi =\frac{180{}^{\circ}-H}{180{}^{\circ}}\times 100\%$$
where *H* (°) was the full width at half maximum of the azimuth scan profile of the Debye ring in the equatorial direction.

The interplanar spacing (*d*) of PA1010 was calculated by the Bragg Equation (3) [30]:(3)2dsinθ=nL,n=1,2⋅⋅⋅
where *θ*, *L* and *n* was the diffraction angle, the wavelength of the X-ray and the diffraction order, respectively.

The Young’s modulus (*E*) was obtained by Equation (4) [41]:(4)$$E=\frac{3F}{\left(\lambda -\frac{1}{\lambda^{2}}\right)}$$
where *F* and λ was the tensile force and draw ratio, respectively. Actually, the dimensional deformation and changes of viscous region were not considered.

## 3. Results and Discussion

### 3.1. Structure of γ-PA1010

The XRD patterns of the PA1010 under different annealing temperature are shown in Figure 1. Two reflection spots with 2θ-angles of 7.34° and 20.84°—corresponding to the *d*-spacing of 1.2034 and 0.4259 nm are observed at RT, which can be indexed as (002) and (100) crystal planes from γ-crystal structure, respectively [38,42,43]. In fact, the reflection spot (100) is the result from the (100) and (010)/(110) overlapped each other, just called reflection spot (100) [29,35]. Generally speaking, during the annealing process of polymers, the energy absorbed by the molecular segments will increase and the molecular segments will undergo creep, rotational and vibrational movements, causing the shift of reflection spots. The reflection spot (100) has shifted slightly angle to right and the intensity increases as the increase of annealing temperature. Compared with the RT, the reflection spot (002) has shifted 0.5°–0.8° to the right under high annealing temperatures and the *d*-spacing becomes smaller, but the intensity has almost not changed, indicating that the molecular segment has undergone torsional motion, even maybe accompanied by rearrangement. The PA1010 hold two reflection spots, which indicates that the γ-crystal structure of the melt-quenched PA1010 could also exist stably under high annealing temperatures. It also shows that the high annealing temperatures (100–140 °C) could not promote the Brill transition of the γ-PA1010.

### 3.2. Structural Evolution of -PA1010

WAXD curves and the corresponding 2D WAXD patterns of the γ-PA1010 stretched with different ratios at RT are shown in Figure 2 and Figure 3, respectively. The *d*-spacing and Full Wave at Half Maximum (FWHM) of (002) of the γ-PA1010 stretched with different ratios at RT are summarized in Figure 4, respectively. The intensity of the reflection spot (100) remains basically unchanged when the draw ratio is less than 2.00 times and then gradually decreases when the draw ratio over 2.00 times, whereas the intensity of the reflection spot (002) gradually increases with the increase of draw ratio. When the draw ratio reaches about 2.00 times, the reflection spot (002) becomes sharper and both the 2θ angle and FWHM of reflection spot (002) gradually decrease, suggesting the increase of *d*-spacing and orientation, respectively (Figure 3). As we all known, the (002) crystal plane is parallel to the X-axis (a-axis) and Y-axis (b-axis) and cut the crystal plane of the repeat unit of the molecular chain. There is only one molecular chain in the unit cell of γ-PA1010, which lacks changes of molecules surface in these directions and thus lacks altering of intermolecular distances. The (002) *d*-spacing gradually increases, indicating that the molecular chain of γ-PA1010 straightens and the (002) crystal plane rearrange along the molecular chain direction.

XRD patterns and in situ 2D WAXD patterns of the simultaneous thermal stretched PA1010 with different ratios at higher temperature (take 120 °C as an example) are shown in Figure 5 and Figure 3 and the *d*-spacing of each reflection spot of γ-PA1010 in Figure 6. The α-crystal of PA1010 belongs to the triclinic system (a = 0.49 nm, b = 0.54 nm, c = 2.782 nm, α = 49°, β = 77° and γ = 63.5 °) [43,44]. There are two main crystal reflection spots, which are (100) and (010)/(110), respectively. The γ-crystal of PA1010 belongs to the monoclinic system, which has only one reflection spot (100). In fact, γ-crystal of PA1010 is a pseudo-hexahedron structure [43,44]. The 2θ angle of the reflection spot (002) gradually decreases with the increase of stretched ratio, meaning the increase of *d*-spacing. As shown in Figure 4, the peak fitting is performed when the draw ratio is 2.00 times. The 2θ angles of the two new reflection spots are respectively 20.82° and 21.62°, corresponding to (100) and (110)/(010) of the α-crystal of PA1010 [45,46,47]. It shows that the α-crystal structure is formed in the course of simultaneous thermal stretched γ-PA1010, and the α-crystal and γ-crystal structure of the PA1010 exist simultaneously. The *d*-spacing of the reflection spot (100) increases first and then decreases, reaching the maximum data when the draw ratio is about 2.00 times. But the new reflection spot (110)/(010) generally increases and then disappears when the draw ratio is 2.50 times in Figure 5, implying that the α-crystal structure of PA1010 has completely transformed into the γ-crystal structure when the draw ratio is 2.50 times and the Brill transformation is completed. This indicates that γ-crystal of PA1010 can induce crystallization and crystal transition in the course of simultaneous thermal stretching.

### 3.3. Crystallinity of γ-PA1010

The crystallinity of γ-PA1010 under different temperature is illustrated in Figure 7. At the same stretched ratios, the crystallinity of γ-PA1010 at RT is obviously lower than that at high annealing temperature, indicating that the high annealing temperature could promote the crystallization of γ-PA1010. The α-crystal structure forms in the course of simultaneous thermal stretched γ-PA1010 with the different higher temperature, resulting in the (110)/(010) and (100) separated in Figure 5. The crystallinity just increases 2–4% and only a small amount of α-crystal form at RT in Figure 6, confirming that the *d*-spacing of the reflection spot (100) increases but not obviously separated (Figure 1).

The crystallinity will increase along with the increase of draw ratio and then decrease when the draw ratio is over 2.00 times. The crystallinity of γ-PA1010 reaches the maximum data when the draw ratio is about 2.00 times. The tensile force can promote PA1010 orientation (Detailed in Section 3.4) and form ordered regions, accelerating the crystallization rate. In addition, the stretching causes the partial fracture of hydrogen bond between the carbonyl group and the amino group of the adjacent polyamide chain and results in a small amount of free amino groups [48]. The weakening of hydrogen bonding reduces the restriction of hydrogen bonding on methylene segments, facilitates the movement of molecular chains and promotes the occurrence of stretch-induced crystallization and crystal form transformation [49,50]. These two factors result in the increased crystallinity. The crystallinity of the PA1010 gradually decreases after the draw ratio reaches 2.00 times, which maybe the tensile force destroyed the crystal structure. We will focus on this phenomenon in future research. It indicates that the behavior of stretch is the principal factor for the crystallinity of γ-PA1010 and the higher temperature annealing is the secondary factor in the course of the synchronous thermal stretched γ-PA1010.

### 3.4. Orientation of the γ-PA1010

The 2D WAXD patterns of γ-PA1010 at RT and 120 °C are shown in the Figure 3 and the reflection spots on the equator and meridian are (100) and (002), respectively. The data of the orientation are summarized in the Table 1. The relationships between the degree of orientation and the draw ratios in the course of synchronous thermal stretched γ-PA1010 can be visually observed. When the draw ratio reaches 1.50 times under higher temperature, the reflection spot (100) is no longer diffuse ring and has some certain radian, at the same time, the reflection spot (002) on the meridian gradually becomes blurred and eventually disappears. As the draw ratios increase, the arc of the reflection spot (100) becomes smaller and the degree of the orientation gradually becomes greater as seen in Table 1. The orientation of the γ-PA1010 gradually increased in the course of synchronous thermal stretching, as shown in Table 1. The degree of orientation of γ-PA1010 is much greater at higher temperature than RT when draw ratio is 1.50 times. Although the difference of the degree of orientation gradually becomes weaker as the draw ratios increases, it is always greater at higher temperature.

During the temperature rise of aliphatic polyamide, the intermolecular hydrogen bonds will become weaker, but the hydrogen bonds do not dissociate [51]. Because PA1010 is a kind of typical aliphatic polyamides, the decrease in hydrogen bonding strength will lead to smaller interactions between the molecular chains of the polyamide, so the molecular segments are easily oriented during the stretching process. When the annealing temperature is higher, the orientation is more easily promoted, indicating that the behavior of simultaneous thermal stretching can promote the high orientation of PA1010.

### 3.5. Mechanical Properties of Synchronous Thermal Stretch γ-PA1010

The relationship between the tensile force and draw ratios in the course of synchronous thermal stretched γ-PA1010 is shown in Figure 8. The relationship between the tensile force and the draw ratio is sublinear at RT, which increases with the increasing of draw ratio. However, there is a linear relationship at higher temperatures between the tensile force and draw ratios and the tensile force increases monotonically with the increasing of draw ratio. The explanation is that the deformation of PA1010 is plastic at RT, whereas PA1010 behaves the elastic deformation at higher annealing temperature.

The relationship between Young’s modulus and draw ratios in the course of synchronous thermal stretched γ-PA1010 is shown in Figure 9. The Young’s modulus of γ-PA1010 gradually decreases but the reduction becomes smaller and smaller with the increase of draw ratios at RT, which corresponds to the mechanical properties of stretched some polymer films [52]. While the Young’s modulus is almost no change in the course of synchronous thermal stretched γ-PA1010 at higher annealing temperature, which may be related to the γ-crystal structure of PA1010 [39]. The Young’s modulus at high annealing temperatures is always smaller than at RT. The young’s modulus decreases with the increase of annealing temperature at the same draw ratio.

## 4. Conclusions

The γ-PA1010 crystal structure was obtained by melt-quenched and could exist stably at 100–140 °C. The tensile force could induce the γ-PA1010 crystalline and form the α-crystal structure in the course of simultaneous thermal stretching. The crystallinity is higher at higher temperature than RT at the same draw ratio. In addition, the crystallinity increases along with the increase of draw ratio (<2.00 times) and then decreases when the draw ratio over 2.00 times. The γ-PA1010 has much greater degree of orientation at higher annealing temperature than RT, but the difference of the degree of orientation gradually becomes weaker with the increase of draw ratio. The effect of tensile force is stronger than the effect of higher annealing temperature on crystallization and orientation during the simultaneous thermal stretched PA1010.

Actually, all the results based on X-ray diffraction represent only the averaged structural behavior in this study and the real microstructure changes may be more complex.

At present, there are relatively few systematic studies on the synchronous thermal stretched polyamides and many issues need to be further considered, such as even higher temperatures and larger draw ratios. More researchers are needed to study continuously and focus more energies on these fields.

## Figures and Tables

**Figure 1 materials-13-01722-f001:**
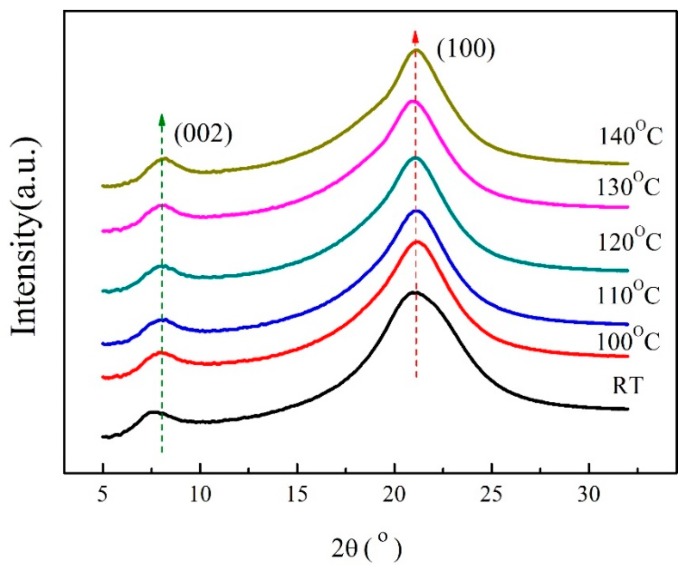
XRD patterns of PA1010 at different annealing temperature.

**Figure 2 materials-13-01722-f002:**
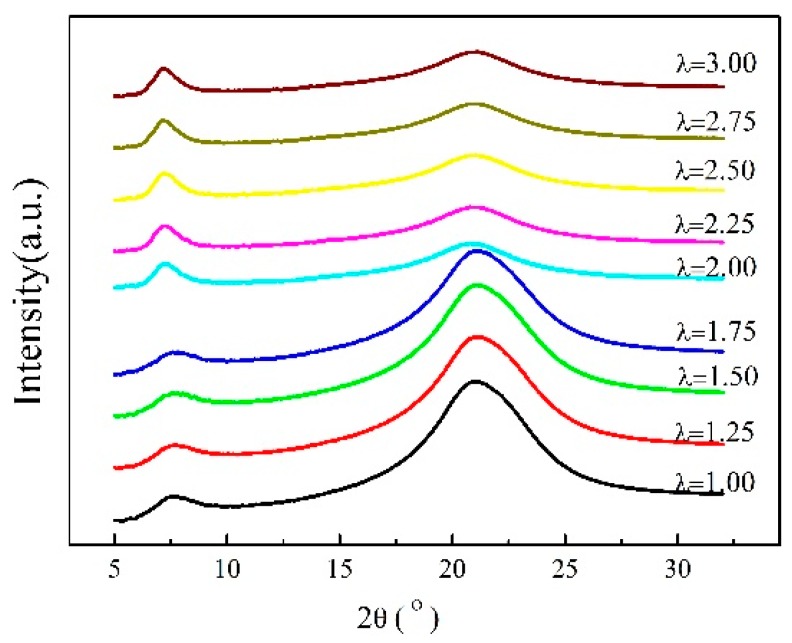
XRD patterns of γ-PA1010 stretched at RT.

**Figure 3 materials-13-01722-f003:**
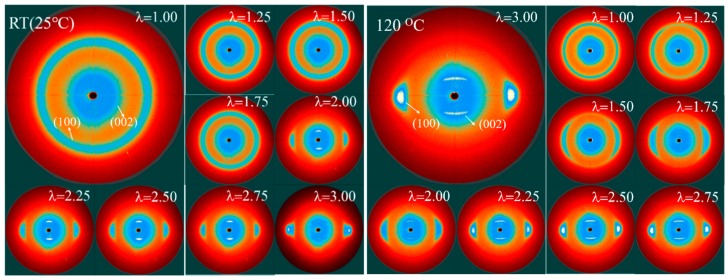
The 2D wide-angle X-ray diffractometer (WAXD) patterns of γ-PA1010 at RT and 120 °C.

**Figure 4 materials-13-01722-f004:**
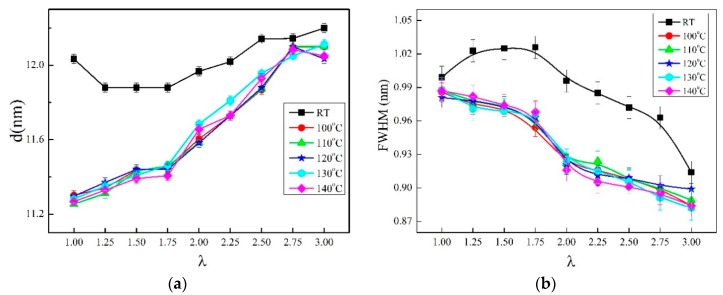
The *d*-spacing and FWHM of (002) of synchronous thermal stretched γ-PA1010 at different temperature. (**a**): *d*-spacing; (**b**) FWHM.

**Figure 5 materials-13-01722-f005:**
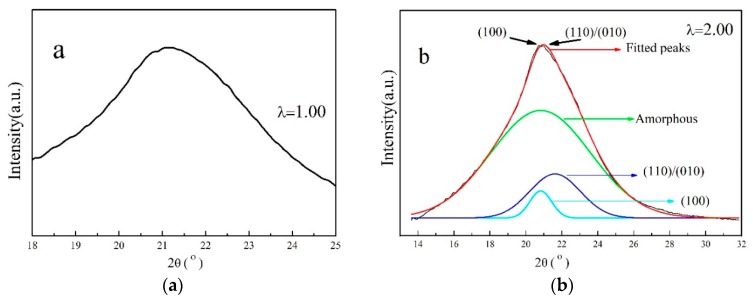
XRD patterns of γ-PA1010 stretched at 120 °C ((**a**–**c**) are partial high magnification image of = 1.00, 2.00 and 3.00, respectively); (**d**) XRD patterns of different draw ratio.

**Figure 6 materials-13-01722-f006:**
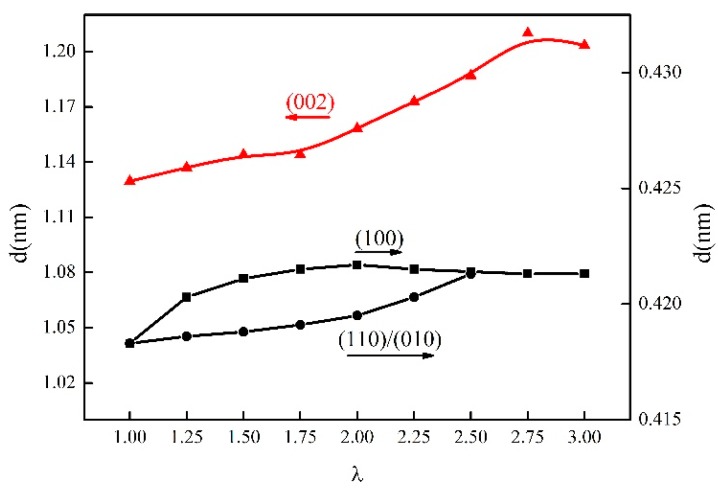
The *d*-spacing of the γ-PA1010 stretched at 120 °C.

**Figure 7 materials-13-01722-f007:**
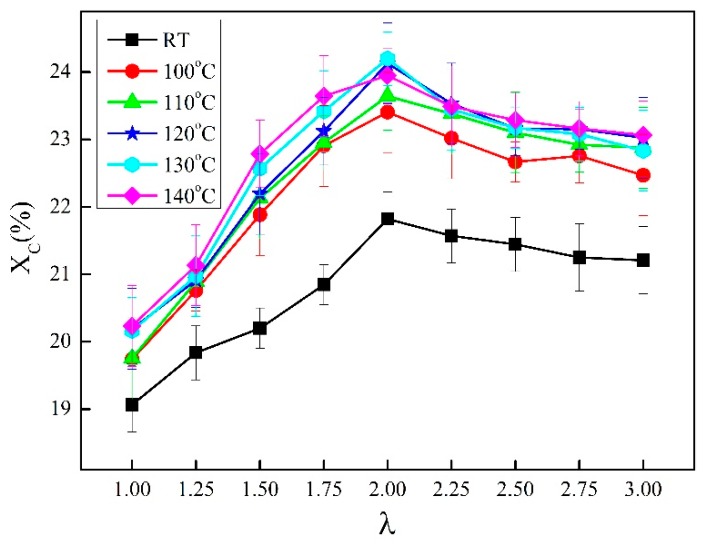
The crystallinity of γ-PA1010 with different temperatures and draw ratios.

**Figure 8 materials-13-01722-f008:**
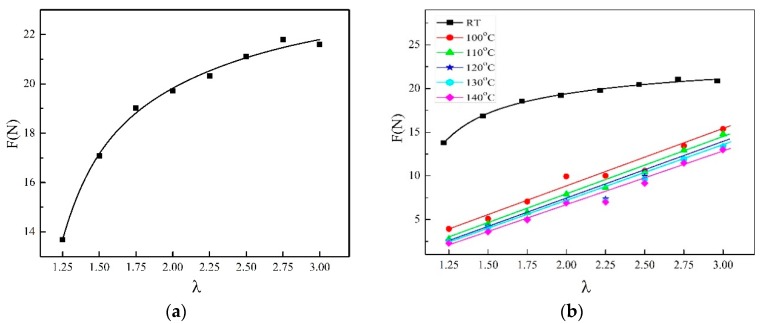
Tensile force of γ-PA1010 with different draw ratios. (**a**): RT; (**b**) different temperature.

**Figure 9 materials-13-01722-f009:**
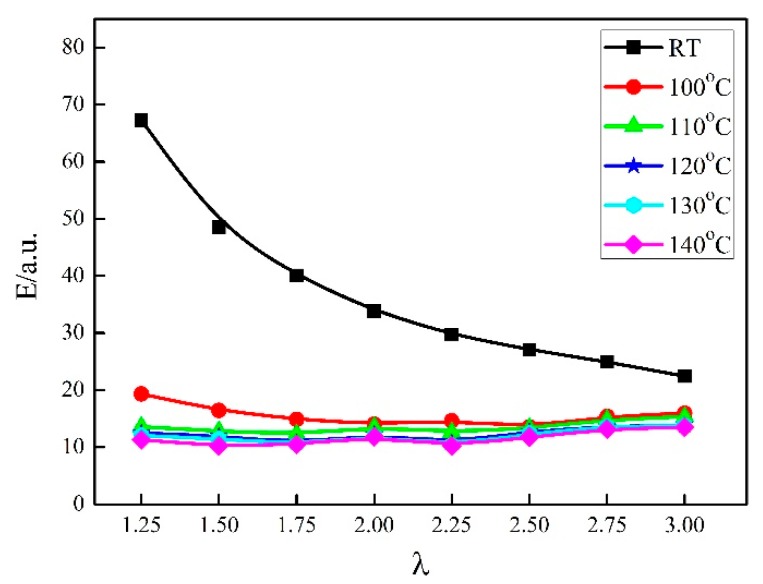
Young’s modulus of γ-PA1010 different draw ratios.

**Table 1 materials-13-01722-t001:** The degree of orientation of reflection spot (100) of synchronous thermal stretched γ-PA1010 at different temperature.

Draw Ratio	Temperature					
	RT (°C)	100 (°C)	110 (°C)	120 (°C)	130 (°C)	140 (°C)
1.50	11.2	44.8	47.8	48.7	48.9	49.2
2.00	56.1	69.5	70.2	72.1	74.1	73.5
2.50	75.4	88.9	89.4	90.1	90.5	90.1
3.00	86.7	91.3	91.9	92.4	93.4	93.3

Note: the value = a ± 4.0, a is the date.

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
