# Peer review of "The Thermal Behavior of γ-PA1010: Evolution of Structure and Morphology in the Simultaneous Thermal Stretched Films"

_materials, 2020, doi:10.3390/ma13071722_

Round 1

Reviewer 1 Report

The authors investigated the crystalline properties of gamma-PA1010 as a function of the temperature and the stretching.

The work seems basically identical to the recent experiment from the same authors presented in Ref 37 on alpha phase. While the investigated phase is different it is unfair to almost perfectly replicate the paper structure, including some parts that are basically a cut and paste (like the experimental).

Moving to more specific comments:

1) The quality of the English is extremely poor, with very numerous grammar errors that makes in general difficult, and sometimes impossible, to understand the meaning of many sentences (for example many sentences do not have the verb). The authors should ask to some English native colleague to revise the manuscript before the resubmission to any journal.

2) The quality of the images is also in general rather poor, making some of them impossible to understand (like Fig. 7 in which the stretch values are too small to be readable).

3) The data shown in Fig. 2 are described concluding that the spectra are almost constant for stretch ratio below 2, and then progressively change for higher values. However the spectra basically show a drastic variation between 1.75 and 2, and then looks pretty similar. Whatever is happening in the sample it seems to be not gradual.

4) The data of the (100) peak at 120° are discussed speaking of a "peak splitting" that is however basically not present. The peak present a shoulder, but there is no evidence that the shoulder relative intensity is increasing with the stretching. The authors should add some quantative analisys (like a multigaussian fit) before drawing conclusions.

5) It is not clear what is the correlation between the authors conclusion and the shown data. In particular in three points the authors report very detailed description of the sample evolution, that are not evident to correlate to the data. For example all the description in lines from 162 to 172,  from 192 to 200 and from 204 to 216 are highly speculative. Moreover  I have not clear why the authors finds surprising the linear dependence between stretch force and stretch ratio, that looks pretty normal in case of elastic deformation. On the contrary the data ("dates" refers to the calendar, the numeric results are "data") at room temperature are typical of anelastic behavior. Thus it simply seems that the sample deformation is plastic at RT and elastic at higher temperatures. I thus do not understand all the discussion from row 204 to row 216, in which crystallization, orientation, energy consumption are considered.

6) The Young modulus makes sense only if the force-deformation relationship is linear. Thus its decrease at RT simply reflects the non linearity of the plot.

Overall the authors performed many measurements on the gamma phase of PA1010, but failed to clearly describe their results and also to convincingly correlate the data to the conclusions. I thus find the manuscript not suitable for publication in Materials.

Author Response

Dear reviewer :Please see the attachment. Thank you!

Reviewer 2 Report

The paper by Zhang et al. addresses the structural development of long-chain polyamide PA1010 under stretching conditions. For this purpose the authors utilise the technique of WAXD together with a hot stretching stage.

The PA101 exists in a pseudo-hexagonal γ-phase after melt-quenching. The data in Fig. 1 support the notion that this phase is maintained at all annealing temperatures between 100-140 °C, in the absence of stretching. However, when stretching is combined with annealing, critical changes in the relatice intensities of the (002) and (100) peaks take place at λ≈2 (Fig. 2). The range of annealing temperatures chosen also leads to a reduction of the (002) d-spacing compared to room temperature, followed by a systematic increase towards the room-temperature d(002)-value with stretch ratio up to 3 (Fig. 3).

With respect to the (100) peak, a splitting is proposed between a (100) and a (100)/(0100) peak,supported by the data in Figs. 4 and 5. Reference to an (0100) peak is incorrect, as this contradicts the Miller-Bravais indexing system. In L. 135 they incorrectly refer to the (110)/(010) peak as characteristic of the alpha-phase, whereas in fact it belongs to the pseudo-hexagonal phase. In L. 136 they write: "This shows that the alpha-crystal structure was formed during the process of γ-PA1010 simultaneous thermal stretchung". The whole discussion between L. 131 and 145 is confused and cannot be published in this form. The indexing of the peaks between the gamma- and alpha-phases is not defined, and they should define what they mean by the alpha-phase. The autors use the term "Brill temperature" here to refer to a transition between the gamma- and the alpha-phase upon stretching and simultaneous annealing.

The discussion of crystallinity-development in LL. 160-172 is plausible, as is the conclusion that stretching is the principal factor governing this, compared to the annealing temperature.

The experimental data pertaining to the other issues, orientation and mechanical properties, as represented in Fig. 8 and 9, are plausible. The straight-line fit to the room-temperature data in Fig. 8 is unacceptable. The discussion is imprecise, this also being a consequence of the lack of clarity regarding the proposed gamma-to-alpha Brill transition.

It would be helpful, in connection with eq. (4) in L. 87, that the reader be remined that λ is the stretch ratio, as in the previous equation, eq. (3), it refers to the wavelength of the X-radiation.

My overall recommendation is that the article should not be published in the current form. The authors should pay careful attention to the indexing of the peaks in Figs. 4 and 5, and significantly improve the clarity of their interpretation of the results. The article is particularly deficient in LL. 131-145. If it is not possible to assign unambiguous phases to the WAXD data, the authors should state this and merely present their experimental results as a novel contribution to the field.

The English is often deficient. However, a competent copy editor would be able to improve this. Scientific points connected with the English are listed below.

Detailed comments on scientific English:

L. 35. It is likely that the authors mean "precession" instead of "procession".

L. 94. "with 2θ-angles" should be written instead of "where the deposition".

Author Response

Dear reviewer:Please see the attachment. Thank you!

Reviewer 3 Report

Review

The submitted manuscript is an extension of authors’ scientific paper published in Journal of Polymer Research, 2019, 26(12):284 (references no 37 of current manuscript). Unfortunately, authors divided one narrow scientific topic into two papers instead of publishing of whole data in one work. Nevertheless, currently presented data should be published as the previously published paper presents incomplete research and consequently the discussed outcomes are limited. Because manuscript possesses some errors (described below), the reviewer cannot recommend publication of the manuscript in the current form and suggests revision.

Comments

The reference to closely related scientific paper (published by Ziqing Cai, et al., in Polymer, Volume 117, 19 May 2017, Pages 249-258, https://doi.org/10.1016/j.polymer.2017.04.037, The structure evolution of polyamide 1212 after stretched at different temperatures and its correlation with mechanical properties) should be added.

The following paper related to studied topic: “Peng Chen, et al., Polymer Testing 67, March 2018, DOI: 10.1016/j.polymertesting.2018.03.035, Systematical mechanism of Polyamide-12 aging and its micro-structural evolution during laser sintering” should be also referred in the manuscript.

The language of manuscript and chemical terminology should be corrected. In current form in some parts it is partially incomprehensible. A few examples of incorrect phrases (of multiple exiting in whole text) present in Paragraph 2: line 64 “with the weight average molecular weight is” (should be “with the average molecular weight equal to ”) , line 68 “all specimens were performed using in situ wide-angle X-ray diffraction” (should be “all specimens were measured with use of in situ wide-angle X-ray diffraction”) line 70 “The specimens were annealed at 100°C, 110°C, 120°C, 130°C and 140°C, and simultaneously the velocity of stretched is 10 μm/s.” (mismatch of tenses and sentence construction error), “The crystallinity (Wc,x) was obtained by the equation (1), which calculated the intensity of diffraction peaks in the crystal and amorphous regions of PA1010.” (unclear, probably should be: The crystallinity (Wc,x) was calculated according to the equation (1), on the basis of area of diffraction spots (of crystalline phase) and area of amorphous phase scattering PA1010.”. Please note that terms “crystalline phase“ and “crystal” have different meaning (and they are incorrectly used in some parts of text). The term “diffraction peak” is incorrect (nevertheless of its common usage) and must be replaced by term “reflection spot” (or eventually by word “reflection”).

The sharpening of reflection spots (described e.g. in line 114) must be numerically represented by values of full width at half maximum (FWHM).

The sentence about blockage of crystallographic c axis change (in line 118) by hydrogen bonds is incorrect as hydrogen bonds at studied conditions possess insignificant strength and cannot be responsible for observed effect. This originates probably from the lack of changes of molecules surface in these directions and thus lack of altering of intermolecular distances.

The texturization of material should be rather represented by pole figures that by crude diffraction patterns (Fig 7). In current form (presented in Figure 7) it is hard to see the differences as two used colours do not allow seeing of the reflection spots border. Additionally, the labels in Figure 7 are too small and possess too low contrast to be readable.

The presented stretch force of synchronous thermal stretch (with stretch ratios; Figure 8) show considerable nonlinearity, especially for the room temperature. It must be studied further, i.e. authors must assume nonlinear function or prove that measured values possess errors within which function is liner (by performing the experiments in multiplicate and subsequent analysis of variance).

Authors should also prove repeatability of registered values, as studied samples are not completely identical (due to character of sample itself) and thus some deviation for repeated measurement must exist (maybe these deviations are unimportant in range of reported values, but it still should be experimentally proven).

The SEM micrographs should be registered for samples before and after experiments as presented results based on X-ray diffraction represents only the averaged structural behaviour. The registration of SEM micrographs allows evaluation of the uniformity and topology of microstructure changes.

The Figures 1 and 2 should be combined into one figure.

The Figures 3 and 5 should be combined into one figure.

The Figure 4 must have the assumed arbitrary units (to allow comparison of all graphs).

The Figures 8 and 9 should be combined into one figure.

Author Response

Dear reviewer: Please see the attachment. Thank you!

Round 2

Reviewer 1 Report

The authors revised the manuscript with the aim of addressing my comments.

Overall the quality of the manuscript and of the date presentation and discussed is strongly improved. While I appreciate the efforts of the authors I still suggest to address the following (minor) comments:

1) The English has been revised, but there are still many mistakes. For example the first sentence of the abstract does not have any verb " In this work, the polyamide 1010(PA1010)prepared by melt-quenched". In row 33 "play a decisive effect " should be "play a decisive role" or "have a decisive effect".  In general a large fraction of sentences still have clear errors, line missing subject or verb. As there are also many correct sentences the authors should make a further effort to correct all the remaining errors.

2) In the introduction the authors write that the research has been mainly focused on short PA. It would be helpful to introduce the particular interest for PA1010 that they are investigating.

3) The authors modified the discussion of the WAXD spectra. However the discussion of the spectra in Fig. 2  again writing that the spectrum evolution between stretching 1.75 and 2 is gradual. As they have an almost stretching independent spectrum up to 1.75 and a stretching independent spectrum from 2, and no intermediate measurements it's hard to conclude that the evolution is "gradual". A gradual evolution is instead present in the spectra of Fig. 4.

4) The stretch force dependence at room temperature in Fig.8 is increasing but it clearly cannot be a parabola. The increase is sublinear, similar to a square root, but anyway unless the authors make a fit with a power low to determine the exponent they can just conclude that the increase is sublinear. In line 211 I suggest to add "at higher temperatures" after "linear increase", otherwise the sentence contradicts the previous one.

Author Response

Dear review: please see the attachment. Thank you!

Reviewer 2 Report

I have read the authors' responses to the points raised and examined the changes made to the manuscript.

Significant improvements have been made and I recommend that the work now proceed to publication.

The intensive input of a copy-editor is required, in order to make the article understandable for a wide audience.

Author Response

(The authors gave the same response as above.)

Reviewer 3 Report

Review

In revised manuscript authors introduced alterations according to the most of reviewer comments. In one case (comment about registering of the SEM micrographs for samples before and after experiments (to evaluate the uniformity and topology of microstructure changes)) authors state that they are not able to register SEM micrographs due to current problems. It is an unfortunate situation, but authors should at least add the comment to manuscript that results based on X-ray diffraction represents only the averaged structural behaviour and real microstructure changes might be more complex.

Authors correct partially the language of manuscript, but it still requires extensive corrections. Authors should ask fluent English speaker (familiar with chemistry) for help or should consider the use of MDPI English Editing Services.

Author Response

(The authors gave the same response as above.)
